# Prognostic features of endometrial cancer metastasis to the central nervous system

**Michelle L. Kuznicki**[1,2¤a], **Adrianne Mallen**[1,2¤b], **Kristal Ha**[2], **Emily Clair McClung**[1,2¤c], **Antonio V. Castaneda**[3,4¤d], **Biwei Cao**[5], **Brooke L. Fridley**[5], **Hye Sook Chon**[1,6], **Jing Yi Chern**[1,6], **Mitchel Hoffman**[1,6], **Robert M. Wenham**[1,6], **Koji Matsuo**[3], **Mian M. K. Shahzad**[1,6]*

1 Department of Gynecologic Oncology, H. Lee Moffitt Cancer Center, Tampa, FL, United States of America, 2 Department of Obstetrics and Gynecology, Morsani College of Medicine, University of South Florida, Tampa, FL, United States of America, 3 Department of Gynecologic Oncology, University of Southern California, Los Angeles, CA, United States of America, 4 Department of Gynecologic Oncology, Ohio State University, Columbus, OH, United States of America, 5 Department of Biostatistics and Bioinformatics, H. Lee Moffitt Cancer Center and Research Institute, Tampa, FL, United States of America, 6 Department of Oncologic Sciences, Morsani College of Medicine, University of South Florida, Tampa, FL, United States of America

¤a Current address: Cleveland Clinic Women's Health Institute, Cleveland, OH, United States of America
¤b Current address: Department of Gynecologic Oncology, Park Nicollet Women's Center, St. Louis Park, MN, United States of America
¤c Current address: Department of Gynecologic Oncology, University of Arizona, Tucson, AZ, United States of America
¤d Current address: Department of Gynecologic Oncology, Hoag Hospital, Orange, CA, United States of America
* Mian.Shahzad@moffitt.org

**Data Availability Statement:** All de-identified patient data has been up loaded as supporting information to PLOS ONE site with this submission.

## Abstract

### Objectives

Central nervous system metastases (CNSm) secondary to endometrial cancer (EC) are rare. As a result, prognostic factors for this patient population are not well described.

### Methods

EC patients with CNSm were identified retrospectively from two academic centers. EC patients without CNSm (non-CNSm) were used as controls. Chi-square and Fisher's exact tests were used for analysis of categorial variables. Wilcoxon tests were used for quantitative measures. Overall survival (OS) was compared with Log-rank test. Cox proportional hazard models were used to estimate hazard ratios for OS.

### Results

22 EC patients with CNSm and 354 non-CNSm patients were included. Compared to non-CNSm EC, the CNSm cohort was younger (58.5 vs 62.0 years, p = 0.018) with lower BMI (27.7 vs. 33.7 kg/m$^2$, p = 0.005), and had more advanced stages (p = ≤ 0.001), grade 3 tumors (81.8% CNSm vs 25.1% non CNSm, p≤0.001) and serous histology (22.7% vs 8.5%, p = 0.010). Median survival after CNSm diagnosis was 9 months (95% CI 4, NA). CNSm was a strong poor prognostic factor (HR death 4.96, p = 0.022). Improved OS was

**Funding:** The author(s) received no specific funding for this work.

**Competing interests:** Honorarium, Chugai, outside the work (KM). Honoraria for advisory board, data safety monitoring committee, steering committee, investigator research support, or speaking from Merck, Genentech, Ovation Diagnostics, GSK/ Tesaro, Clovis, AstraZeneca, Mersana, Abbvie, Legend Biotech, Regeneron, Seagen/Seattle Genetics, outside of the work (RMW).

seen with CNS as the only disease site (83m CNSm only vs 30m additional sites, p = 0.007) and less than five CNSm (49m <5 vs. 23m $\geq$5, p = 0.004). Surgical resection of CNSm (OS 83m surgery vs 33m no surgery, p = 0.003) or multimodal therapy (83m multimodal vs 33m single therapy, p = 0.027) resulted in longer OS.

## Conclusions

CNSm is a poor prognostic factor in EC, however, low volume disease with aggressive treatment may result in more favorable survival outcomes.

## Introduction

Endometrial cancer (EC) is the most common gynecologic cancer with 65,000 new cases estimated in 2020 [1]. EC has traditionally been divided into Type 1 and Type 2 tumors, which is based on histology and correlates with clinical outcomes [2]. Type 1 tumors are comprised of endometrioid histology, usually of low to moderate grade, and tend to occur in the setting of obesity and advanced age with a generally favorable prognosis [3]. Alternatively, type 2 tumors include high grade histologies such as grade 3 endometrioid adenocarcinoma, serous carcinoma, clear cell carcinoma and carcinosarcoma which are more prone to metastasis and recurrence, and therefore considered high risk [2, 4]. This model has proven to be imperfect with certain type 1 tumors, not infrequently, exhibiting aggressive clinical features [2].

Spread of EC from the primary site can occur by way of lymphatics to locoregional lymph nodes, local invasion to surrounding tissues, or hematogenous spread to distant organs [5]. Distant metastasis is an uncommon occurrence, present approximately in only 15% of patients at the time of initial diagnosis [1]. Central nervous system metastasis (CNSm) from EC represents a very small subset of distant metastasis with reported incidence between 0.3 to 1.2% [6]. As expected, CNSm from EC has previously been associated with higher tumor grade, advanced stage and high-risk histologic types [7–9]. Interestingly, CNSm from gynecologic cancers appears to be increasing over time, although, it is thought to be secondary to prolonged survival and advancements in imaging techniques [10]. However, due to the rarity of CNSm from EC, there remains limited data to inform patient counseling and appropriate treatment modalities.

In the present retrospective cohort study, conducted at two major NCI designated cancer centers, we sought to describe characteristics and survival outcomes of patients with CNSm from EC. Additionally, we aimed to evaluate potential prognostic factors associated with improved outcomes following CNSm diagnosis. Due to the rare nature of this presentation, we further aimed to complete a review of the literature and provide additional valuable insight on its outcomes.

## Materials and methods

This retrospective cohort study includes EC patients with CNSm from H. Lee Moffitt Cancer Center (MCC) and University of Southern California (USC). Institutions were chosen due to preexisting collaborative relationships and were the main practicing sites of involved authors. Institutional Review Board approval was obtained at both institutions which each waived the requirement for informed consent given the retrospective nature of this study. Electronic medical record systems then were queried at both institutions to identify cases of EC with CNSm

diagnosed between January 1,1986 (earliest date of scanned records) through June 1, 2016 (initial start date of data collection). Patients with a diagnosis code of EC and billing code for CNS imaging comprised the screening pool. Inclusion criteria required a CNSm to be identified in an official imaging report with or without a confirmatory biopsy. Only carcinomas arising from the endometrium were included in the present study, which was limited to endometrioid, serous, clear cell or carcinosarcoma. Uterine sarcomas were excluded. Patients were excluded if they had a separate non-EC cancer diagnosis, unless the CNSm was biopsy proven EC primary.

Data regarding demographics, clinical characteristics, pathology, treatment data, and survival outcomes of eligible patients were extracted from the medical record. Overall survival (OS) was calculated from date of primary EC diagnosis to date of death or last follow up. Survival after CNSm was calculated from date of CNSm diagnosis to date of death or last follow up. Data from a preexisting database of all EC patients treated at the primary site (MCC) from 2014–2017 was used to create a comparison cohort (non-CNSm). This data set included all stages of EC, all EC histologies and were included regardless of recurrence status. A limited data set was collected for these patients including demographic information, primary disease characteristics, treatment modality and survival outcomes.

Chi-square or Fisher's exact tests were used when comparing categorial clinical characteristics between EC CNSm cohort and EC non-CNSm cohort or site A and B within CNSm cohort, while Wilcoxon tests were used for comparison of quantitative measures. For analysis within CNSm group only, the associations between categorical variables and interested endpoints were evaluated using Chi-squared tests or Fisher's exact tests when the expected frequencies were low. Categorical variable levels for OS were compared using the Log-rank test, and the corresponding Kaplan-Meier curves were generated. For these analyses within the CNSm cohort we did not adjust for covariates due to the limited sample size. For analysis between CNSm and non-CNSm cohorts, Cox proportional hazards regression model was used to estimate hazard ratios (95% confidence interval) for OS with adjustment for clinical covariates. For the analysis using the larger non-CNSm EC cohort, which involved a larger sample size, we were able to fit multivariable models adjusting for stage, histology and age.

A review of the literature on the present topic was performed using Pubmed with search terms "endometrial cancer" or "uterine cancer" and "brain metastasis" or "central nervous system metastasis." Included articles were limited to studies reporting original data describing outcomes specific to patients with CNSm from endometrial cancer. Systemic reviews and articles reporting on multiple cancer types without delineating data specific to uterine cancer patients were excluded. Reference lists from publications of interest were also used as a search method to identify all relevant articles.

## Results

### Patient characteristics

A total of 22 patients with CNSm from EC were identified (MCC Site A n = 11, USC Site B n = 11). Institutional cohorts were comparable for all clinical and pathologic characteristics aside from race where the USC cohort represented the majority of non-white patients (Table 1). For all patients, initial treatment for primary EC was surgery (n = 12, 54.5%), chemotherapy (n = 8, 36.4%), or unknown (n = 2, 9.1%). The majority of patients had advanced stage disease (Stage III/IV: n = 20, 90.9%) and high-grade tumors (grade 3: n = 18, 81.8%). Additionally, 27.2% represented high risk histology (serous n = 5, carcinosarcoma n = 1).

In comparison with the non-CNSm EC cohort, the CNSm cohort was found to be significantly younger at time of initial EC diagnosis (58.5 vs 62.0 years, p = 0.018), had lower BMI

**Table 1. Clinical and pathologic characteristics.**

| | Total | Site A | Site B | p value |
|---|---|---|---|---|
| | N = 22 (%) | N = 11 (%) | N = 11 (%) | |
| Age at EC Diagnosis | 58.5 (52.2; 61.8) | 56 (52; 64) | 59 (52.5; 60.5) | 0.92 |
| BMI | 27.7 (25; 31) | 27.6 (25.5; 29.6) | 28.4 (24.9; 32.9) | 0.71 |
| Race | | | | 0.001 |
| White | 10 (45.5) | 9 (81.8) | 1 (9.1) | |
| Black | 1 (4.6) | 0 (0) | 1 (9.1) | |
| Hispanic | 4 (18.2) | 2 (18.2) | 2 (18.2) | |
| Asian | 1 (4.6) | 0 (0) | 1 (9.1) | |
| Other | 6 (27.3) | 0 (0) | 6 (54.6) | |
| Initial Stage* | | | | 0.52 |
| I | 1 (4.6) | 1 (9.1) | 0 (0) | |
| II | 1 (4.6) | 0 (0) | 1 (9.1) | |
| III | 10 (45.5) | 6 (54.5) | 4 (36.4) | |
| IV | 10 (45.5) | 4 (36.4) | 6 (54.5) | |
| Initial Grade | | | | 0.72 |
| 1 | 2 (9.1) | 1 (9.1) | 1 (9.1) | |
| 2 | 2 (9.1) | 0 (0) | 2 (18.2) | |
| 3 | 18 (81.8) | 10 (90.9) | 8 (72.7) | |
| Histology | | | | |
| Endometrioid | 16 (72.7) | 8 (72.7) | 8 (72.7) | |
| Serous | 5 (22.7) | 3 (27.3) | 2 (18.2) | |
| Carcinosarcoma | 1 (4.6) | 0 (0) | 1 (9.1) | |
| LVSI | 7 (31.8) | 6 (54.5) | 1 (9.1) | 0.13 |
| CNSm as First Recurrence | 8 (36.4) | 5 (45.5) | 3 (27.3) | 0.66 |
| Additional Disease at CNSm Dx | 15 (68.2) | 7 (63.6) | 8 (72.7) | 0.99 |
| Presenting Symptom of CNSm | | | | |
| Headache/Dizziness | 9 (40.9) | 7 (63.6) | 2 (18.8) | 0.08 |
| Altered Mental Status | 3 (13.6) | 2 (18.2) | 1 (9.1) | 0.99 |
| Nausea/Vomiting | 4 (18.2) | 2 (18.2) | 2 (18.2) | 0.99 |
| Weakness/Numbness | 4 (18.2) | 3 (27.3) | 1 (9.1) | 0.59 |
| Coordination/Gait Issues | 5 (22.7) | 5 (45.5) | 0 (0) | 0.04 |
| Dysphagia | 2 (9.1) | 0 (0) | 2 (18.2) | 0.48 |
| Number of CNSm** | | | | 0.60 |
| 1 | 10 (45.4) | 6 (54.5) | 4 (44.4) | |
| 2–5 | 4 (18.2) | 3 (27.3) | 1 (11.1) | |
| >15 | 6 (27.3) | 2 (18.2) | 4 (44.4) | |
| Treatment Approach to CNSm | | | | 0.21 |
| Surgical Resection alone | 1 (4.6) | 0 (0) | 1 (9.1) | |
| Surgical Resection + localized RT | 5 (22.7) | 3 (27.3) | 2 (18.2) | |
| Surgical Resection + WBRT | 1 (4.6) | 0 (0) | 1 (9.1) | |
| Surgical Resection + Chemo Wafers | 2 (9.1) | 2 (18.2) | 0 (0) | |
| Stereotactic Radiosurgery alone | 3 (13.6) | 3 (27.3) | 0 (0) | |
| No treatment | 3 (13.6) | 1 (9.1) | 2 (18.2) | |

Continuous variables presented as mean (standard deviation). Significant p value considered <0.05.

Abbreviations: EC endometrial cancer, BMI body mass index, LVSI lymphovascular space invasion, CNSm central nervous system metastasis, Dx Diagnosis, f/b followed by, RT radiation, WBRT whole brain radiation therapy, chemo wafers: chemotherapy wafers placed in surgical bed.

*One patient with unknown stage.

** number of CNSm not available for 2 patients.

(27.7 vs. 33.7 p = 0.005) and were more likely to be non-white (p < 0.0001). Advanced stage disease (stage III/IV 86.3% vs 16.4%, p ≤ 0.001), high-grade tumors (81.8% vs 25.1%, p ≤ 0.001) and serous histology (22.7% vs 8.5%, p = 0.01) were all more prevalent in the CNSm cohort (**Table 2**). An additional comparison was completed between the CNSm cohort and patients within the non-CNSm cohort who developed recurrences (n = 30), in attempt to capture the more aggressive cases. In this comparison, the CNSm cohort and the recurrent non-CNSm cohort had similar rates of serous histology (22.7% CNSm vs 23.3% recurrent non-CNSm, p = 0.205) and grade 3 tumors (81.8% CNSm vs 63.3% recurrent non-CNSm, p = 0.370). However, the CNSm cohort continued to have younger age at diagnosis (58.5 years CNSm vs 62.5 years recurrent non-CNSm) and higher rates of stage 3 or 4 disease at time of diagnosis (86.4% CNSm vs 56.6% recurrent non-CNSm, p = 0.029).

## CNS metastasis

In EC patients with CNSm, the median interval between primary EC treatment completion and CNSm diagnosis was 17.5 months (IQR 1.8–26). Routine CNS imaging to screen for CNSm during treatment or surveillance of EC was not a regular practice at either institution in the absence of symptoms. The majority of patients (82%) were diagnosed following new onset of CNS symptoms (**Table 1**). Diagnosis was made by magnetic resonance imaging (MRI) in

**Table 2. Endometrial cancer patients with CNS metastasis compared to control cohort without CNS metastasis.**

|  | CNSm (N = 22) | Non CNSm (N = 354) | p value |
|---|---|---|---|
| Age at dx (years) | 58.5 [52.2;61.8] | 62.0 [55.0;68.8] | 0.018 |
| BMI (kg/m$^2$) | 27.7 [24.9;31.0] | 33.7 [27.3;40.6] | 0.005 |
| Race/Ethnicity |  |  | <0.001 |
| White | 10 (45.5%) | 305 (86.2%) |  |
| Black | 1 (4.6%) | 22 (6.2%) |  |
| Hispanic | 4 (18.2%) | 5 (1.4%) |  |
| Asian | 1 (4.6%) | 7 (2.0%) |  |
| Unknown/Other | 6 (27.3%) | 15 (4.24%) |  |
| Histology |  |  |  |
| Endometrioid | 16 (72.7%) | 262 (74.0%) | 0.043 |
| Serous | 5 (22.7%) | 30 (8.5%) |  |
| Stage* |  |  |  |
| I | 1 (4.6%) | 272 (76.8%) | <0.001 |
| II | 1 (4.6%) | 24 (6.8%) |  |
| III | 10 (45.5%) | 45 (12.7%) |  |
| IV | 10 (45.5%) | 13 (3.7%) |  |
| Grade* |  |  |  |
| 1 | 2 (9.1%) | 156 (44.1%) | <0.001 |
| 2 | 2 (9.1%) | 109 (30.8%) |  |
| 3 | 18 (81.8%) | 89 (25.1%) |  |
| LVSI* |  |  |  |
| Yes | 7 (31.8%) | 98 (27.7) | <0.001 |
| No | 7(46.4%) | 248 (70.1%) |  |
| Not Reported | 8 (36.4%) | 8 (2.3%) |  |

Abbreviations: dx diagnosis of endometrial cancer, BMI body mass index, CNSm central nervous system metastasis, LVSI lymphovascular space invasion.

Continuous variables presented as mean [Interquartile range/IQR]

*Denotes characteristics of primary endometrial tumor

45.4%, computerized tomography scan (CT) in 27.3%, and both MRI and CT in 9.1%. The number of CNSm varied with solitary metastasis identified in 10 patients (45.5%), 2–5 metastases in 4 patients (18.2%), and diffuse metastases (>15 metastases) in 6 patients (27.3%). Location of CNSm included cerebrum alone (31.8%), cerebellum alone (31.8%), both cerebrum and cerebellum (27.3%), and meningeal involvement (4.5%). Diameter of the largest CNSm per patient varied including <1 cm (n = 1, 4.6%), 1–3 cm (n = 7, 31.8%), and 3–5 cm (n = 10, 45.5%), data on largest diameter CNSm was missing for four patients.

## Treatment of CNS metastasis

Treatment modalities for CNSm are displayed in **Table 1** which included whole brain radiation (WBRT) alone (n = 7, 31.8%), surgical resection followed by localized radiation (n = 5, 22.7%), stereotactic radiosurgery alone (n = 3, 13.6%), surgical resection alone (n = 1, 4.6%), surgical resection followed by WBRT (n = 1, 4.6%), and surgical resection followed by placement of chemotherapy wafers (n = 2, 9.1%). Three patients (13.6%) received no therapy for CNSm. Six patients (27.3%) who received treatment for CNSm developed recurrence/progression in the CNS following therapy. Of these six patients, three underwent stereotactic radiosurgery alone, two underwent surgical resection followed by radiation to surgical bed, and one underwent surgical resection followed by placement of chemotherapy wafer.

Treatment modality was found to be associated with number of CNSm. Surgical resection occurred more often in patients with solitary brain metastasis versus multiple metastases (77.78% vs 20%, p = 0.023). WBRT was administered more often in women with greater than five CNSm (100% ≥5 vs 16.7% <5, p = 0.0015). Additionally, multimodal therapy was used more frequently in women with solitary brain metastases compared to multiple (70% vs 8.3%, p = 0.005).

## Survival

OS varied widely in the CNSm EC cohort with a median of 49 months (95% CI: 30, NA). Median survival after CNSm diagnosis was 9 months (95% CI 4, NA). Five patients (22.7%) survived longer than 1 year after CNSm diagnosis (72, 14, 18, 41, 37 months).

We identified improved OS when CNS was the only site of metastasis compared to those with additional non- CNS metastasis (83m CNSm only vs 30m additional sites, p = 0.007) (Table 3). Those with CNSm as the only site of disease also had improved survival following CNSm diagnosis compared to those with concurrent disease outside CNS (p = 0.0096) (**Fig 1.1**). Out of seven patients with CNSm and no evidence of disease outside of the CNS, only one died during the follow-up period. Those with solitary CNSm had improved OS compared to those with multiple CNSm (83m solitary vs 28m multiple, p = 0.04) (Table 3). Patients with

**Table 3. Differences in overall survival based on number of CNSm and disease burden.**

| Comparison Groups | Median Overall Survival | P Value |
|---|---|---|
| CNSm only site of metastasis | 83 months | 0.007 |
| CNSm + other sites of metastasis | 30 months | |
| Solitary CNSm | 83 months | 0.040 |
| Multiple CNSm | 28 months | |
| <5 CNSm | 49 months | 0.005 |
| ≥5 CNSm | 23 months | |

*Abbreviations*: CNSm: central nervous system metastasis, +: in addition to, <: less than, ≥:greater than or equal to

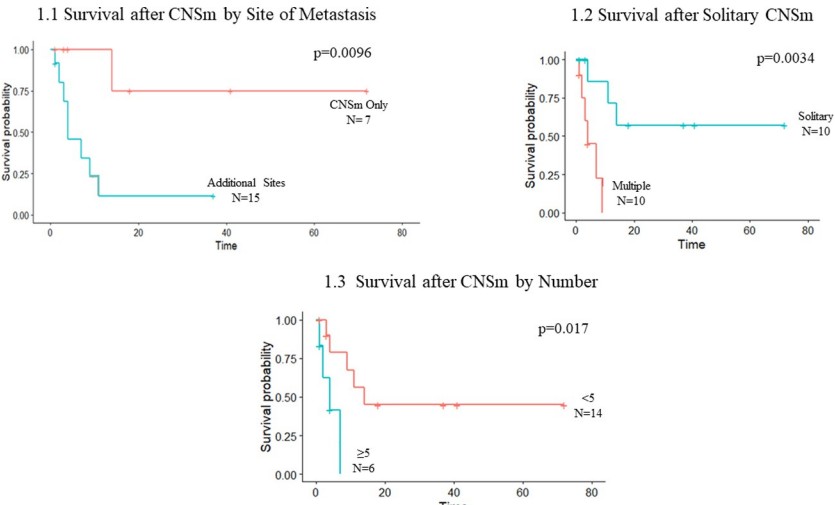

**Fig 1. Survival based on central nervous system metastasis characteristics.** Survival time displayed in number of months. (1.1) Survival following CNS metastasis with CNS metastasis as only site of recurrence versus those with multiple sites of disease. (1.2) Survival following CNS metastasis diagnosis in patients with solitary CNS metastasis compared to those with multiple CNS metastases. (1.3) Survival following CNS metastasis diagnosis in patients with less than five CNS metastases compared to those with five or more CNS metastases.

solitary CNSm also had improved survival following CNSm diagnosis (p = 0.0034) (**Fig 1.2**). Overall, a small number of CNSm was associated with improved OS. OS for patients with less than five CNSm was more favorable compared to those with five or greater CNSm (49m <5 vs. 23m ≥ 5, p = 0.0045) (**Table 3**). Similar survival benefit was found for survival following CNSm diagnosis (**Fig 1.3**). No significantly associated survival benefit was found with increasing diameter of CNSm (6.5months <3cm vs 14months >3cm, p = 0.327).

Compared to the non-CNSm EC cohort, CNSm patients had a significantly higher risk of death (HR 11.3, 95% CI 3.4–36.9, p < 0.001). This remained significant after controlling for stage (HR 4.53, 95% CI1.18, 17.32, p = 0.027), histology (HR 13.51, 95% CI 3.88, 47.04, p = <0.001) or age (HR 10.4, 95% CI 3.18–34.01, p = <0.001). Controlling for all three variables simultaneously resulted in persistent significantly worse prognosis for CNSm (HR death 4.96, p = 0.022). CNSm cohort OS was also compared specifically to patients within the non-CNSm EC cohort who recurred. Although the CNSm cohort trended towards higher risk of death, this was not significant compared to the recurrent patients from non-CNSm cohort (HR 2.28, 95% CI 0.578–8.974, p = 0.239).

An analysis comparing survival outcomes based on treatment approach was completed. Three patients received no treatment for their CNSm, all of whom survived ≤1 month following CNSm diagnosis. Overall survival from EC diagnosis was 29 months, 5 months and 1 month for these three patients. With these three patients excluded, patients who underwent surgical resection of CNSm had longer median OS (83m vs 33m, p = 0.003) and longer median survival after CNSm diagnosis (NR vs. 4m, p = 0.002) compared to those who did not undergo surgical resection of CNSm (**Fig 2.1**). Those who received multimodal treatment for CNSm with any combination of modalities had improved median OS (83m vs 33m, p = 0.027) and improved survival after CNSm diagnosis (NR vs 4m single modality, p = 0.002) (**Fig 2.2**). Additionally, those who received WBRT for treatment of CNSm had decreased OS (23m vs 83m, p = 0.014) and decreased survival following CNSm diagnosis (4m vs 14m, p = 0.022) compared to those who did not receive WBRT (**Fig 2.3**).

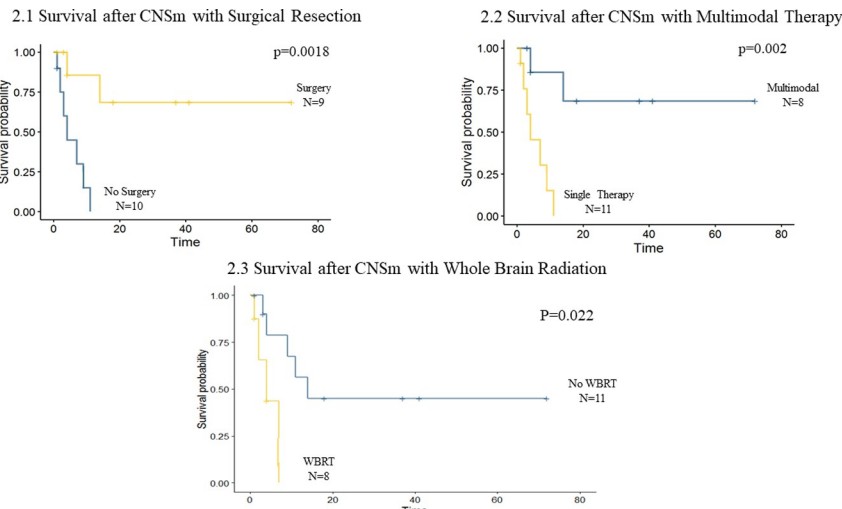

**Fig 2. Survival after central nervous system metastasis based on treatment approach.** Survival after central nervous system metastasis based on treatment approach in endometrial cancer patients with central nervous system metastasis. *Survival time displayed in number of months.* (2.1) Survival following CNS metastasis diagnosis for patients undergoing surgical resection of CNSm versus those who did not undergo surgery. (2.2) Survival following CNS metastasis diagnosis for patients undergoing multimodal therapy for treatment of CNSm compared to those who underwent single modality therapy. (2.3) Survival following CNS metastasis diagnosis for patients undergoing whole brain radiation therapy (WBRT) compared to those who did not undergo WBRT.

## Discussion

Endometrial cancer is described as a "neurophobic disease" that is rarely found to metastasize to the CNS [11]. Neuroimaging is not a routine part of the initial work-up unless there are concerning symptoms. Given the paucity of CNSm cases in EC, it is difficult to identify those at risk, and a diagnosis of CNSm may be delayed due to low suspicion. Additionally, following diagnosis of CNSm it is challenging to appropriately counsel patients on oncologic expectations because prognostic features specific to endometrial cancer are not well-reported. In the present retrospective study, we aimed to describe this unique patient population and evaluate for features associated with development of CNSm and survival thereafter. Multiple risk factors were identified to be significantly more prevalent in EC patients with CNSm compared to those without CNSm including lower BMI, non-white race, advanced stage disease, high-grade tumors, and high-risk histology. Despite the poor prognosis identified in patients with CNSm from EC, the present study found that patients with low disease burden and those who received multimodal therapy had improved survival.

A review of the literature was also completed to correlate our findings compared to those previously described in patients with CNSm from uterine cancers, results summarized in Table 4 [7–10, 12–22]. Fifteen publications, in addition to the present study, were identified describing 274 patients in total, with 240 representing endometrial cancers. Consistent with our results, the majority were advanced stage (Stage III/IV: 189/260, 72.7%) and poorly differentiated tumors (Grade 3: 101/156, 64.7%) with a large representation of high-risk histology endometrial subtypes (56/240, 23.3%) (**Table 4**).

Regarding initial diagnosis of CNSm, prior studies have reported 89–100% of CNS metastasis diagnoses were made following work-up for new neurologic complaints [7, 10, 21]. New neurologic symptoms were present in 83% of patients in the present study at the time of CNSm diagnosis with the most common symptoms being headache and dizziness. Early diagnosis of CNSm following prompt investigation of new neurologic symptoms would potentially

**Table 4. Summary of studies on central nervous system metastasis in endometrial cancer.**

| Author, Year | No. Patients | Histology | Grade | Stage | CNSm Treatment | mOS CNSm |
|---|---|---|---|---|---|---|
| Kuznicki et al, 2021 (Current Study) | 22 | 16 Endometrioid<br>5 Serous<br>1 Carcinosarcoma | 1 : 2<br>2 : 2<br>3 : 18 | I : 1<br>II : 1<br>III : 10<br>IV : 10 | Surgery: 1<br>WBRT: 7<br>Radiosurgery: 3<br>Surgery + localized RT: 5<br>Surgery + WBRT: 1<br>Surgery + chemo wafers: 2 | 4m (1–72) |
| Bhambhvani et al, 2020 [12] | 30 | 16 Endometrioid<br>7 Serous<br>5 Carcinosarcoma<br>1 Glassy Cell<br>1 Clear Cell | 1 : 1<br>2 : 3<br>3 : 24 | I : 6<br>II : 1<br>III : 11<br>IV : 9 | Surgery : 11<br>Radiosurgery : 28 | 6.8m (1–58.2) |
| Ogino et al, 2020 [13] | 12 | * | * | * | Radiosurgery +/- additional treatment: 12 | 4.5m (1–11) |
| Zhang et al, 2019 [9] | 24 | 11 Endometrioid<br>4 Serous<br>3 Carcinosarcoma<br>1 Adenosquamous<br>1 Clear Cell<br>1. Leiomyosarcoma<br>1 Pleiomorphic sarcoma<br>1 Small cell neuroendocrine<br>1 Mixed adenocarcinoma | 1: 2<br>2: 2<br>3: 20 | I: 3<br>II: 2<br>III: 12<br>IV : 7 | * | * |
| Moroney et al, 2019 [14] | 12 | 9 Endometrioid<br>2 Serous<br>1 Adenosquamous | 1 : 1<br>2 : 3<br>3 : 8 | I : 4<br>II : 1<br>III : 3<br>IV : 4 | RT : 4<br>Surgery+ RT : 2<br>RT + chemo : 2<br>Surgery + RT + chemo : 2<br>None: 2 | * |
| Cybulska et al, 2018 [7] | 23 | 23 Endometrioid | 1 : 9<br>2 : 14<br>3: 0 | I : 15<br>II : 2<br>III : 3<br>IV : 3 | RT : 7<br>Surgery + chemo : 2<br>Surgery + chemo + RT: 4<br>Surgery + RT : 3<br>RT + chemo : 1<br>None : 6 | 5.1m (2.2–8.1) |
| Uccella et al, 2016 [21] | 18 | 12 Endometrioid<br>3 Serous<br>1 Adenosquamous<br>2 Undifferentiated | 1 : 2<br>2 : 2<br>3 : 14 | I : 6<br>III : 7<br>IV : 5 | WBRT : 5<br>Radiosurgery : 1<br>Surgery + WBRT : 8<br>WBRT + chemo : 1<br>None : 3 | 6.5m (0–118) |
| Shin et al, 2016 [15] | 6 | 5 Endometrioid<br>1 Small cell Carcinoma | * | * | Radiosurgery +/- additional therapy: 6 | 7.5m |
| Gressel et al, 2015 [10] | 22 | 13 Endometrioid<br>4 Serous<br>2 Carcinosarcoma<br>1 Squamous cell<br>1 Adenosquamous<br>1 Leiomyosarcoma | * | I: 1<br>II: 2<br>III: 6<br>IV: 12 | Surgery : 2<br>RT : 15<br>Surgery + RT: 2<br>None: 4 | 4m (0–123) |
| Kim et al, 2015 [16] | 19 | 6 Carcinoma<br>11 Adenocarcinoma<br>2 Leiomyosarcoma | * | II: 3<br>III: 8<br>IV: 8 | Surgery: 9<br>RT: 14<br>Chemo: 9<br>None: 2 | 23.3m (17.8–28.8) |
| Shepard et al, 2014 [20] | 6 | * | * | I: 2<br>III: 1<br>IV: 2 | Radiosurgery +/- additional therapy: 6 | 8.3m (4–16) |
| Chura, 2007 [17] | 20 | 11 Endometrioid<br>1 Serous<br>2 Adenosquamous<br>3 Carcinosarcoma<br>3 Undifferentiated | 1 : 3<br>2 : 6<br>3 : 11 | I : 3<br>III : 8<br>IV : 9 | WBRT: 7<br>WBRT + Chemo: 4<br>WBRT + Radiosurgery : 1<br>Surgery + WBRT : 1<br>Surgery + WBRT + chemo : 3<br>None : 4 | 2.0m (0.1–39.2) |

*(Continued)*

**Table 4.** (Continued)

| Author, Year | No. Patients | Histology | Grade | Stage | CNSm Treatment | mOS CNSm |
|---|---|---|---|---|---|---|
| Orru, 2007 [19] | 3 | 1 Endometrioid<br>2 Adenocarcinoma, other | 1: 0<br>2: 0<br>3: 3 | III : 3 | WBRT: 1<br>Surgery + WBRT : 2 | * |
| Mahmoud-Ahmed, 2001 [22] | 10 | 7 Adenocarcinoma<br>3 Adenosquamous | * | II : 1<br>III : 4<br>IV : 4 | WBRT: 4<br>Surgery : 2<br>Radiosurgery + WBRT: 1<br>Surgery + WBRT: 2<br>Surgery + radiosurgery + WBRT: 1 | 3.3m |
| Totals | 274 | 147 Endometrioid<br>30 Serous<br>21 Adenocarcinoma NOS<br>18 Carcinosarcoma<br>11 Adenosquamous<br>6 Carcinoma NOS<br>4 Leiomyosarcoma<br>3 Clear cell<br>3 Undifferentiated carcinoma NOS<br>2 Small cell neuroendocrine<br>1 Squamous<br>1 Glassy Cell<br>1 Pleiomorphic sarcoma<br>1 Undifferentiated sarcoma | 1: 20<br>2: 35<br>3: 101 | I: 54<br>II: 17<br>III: 102<br>IV: 87 | | |

Abbreviations: No. number, CNSm central nervous system metastasis, mOS CNSm median overall survival following CNS metastasis diagnosis, RT radiation, chemo chemotherapy, WBRT whole brain radiation therapy

*missing data

mOS : mean/median (range) as reported in the primary reference.

allow for multimodal intervention at a time of lowest volume CNSm [15, 22, 23]. Timing of CNSm diagnosis appears to correlate with timing of disease recurrence with all but four patients presenting with CNSm within three years of original EC diagnosis. Additionally, 36% of patients in the present study were diagnosed with CNSm at the time of their first recurrence. This suggests CNSm can occur early in the course of disease recurrence and CNS imaging should be completed in the setting of new neurologic symptoms regardless of timing from EC diagnosis. Identifying CNSm as early as possible is clinically ideal in order to expedite treatment and avoid potentially catastrophic consequences of untreated CNSm including hemorrhage or herniation. Interestingly, only one patient in the present study was identified as having leptomeningeal metastasis in the absence of a parenchymal lesion. If neurologic symptoms persist in the absence of gross parenchymal metastasis on brain imaging, a consideration may be made for addition of spinal imaging if this has not been completed.

Reported survival following CNSm from EC is poor, consistent with our finding of a median survival after CNSm of 3.5 months (Table 4). We identified CNSm in EC as an independent poor prognostic factor (HR 0.09, p = <0.001) which remained significant after controlling for age, stage, and histology compared to non-CNSm EC patients. Although median survival is not encouraging, there was a wide variability in survival following CNSm diagnosis (range 1–72 months), and there seem to be prognostic features associated with improved outcomes. Additionally, compared to recurrent non-CNSm EC patients, we were unable to demonstrate a significant difference in survival for CNSm patients (HR 2.28, 95% CI 0.578–8.974, p = 0.239). Volume of disease appears to be a driving factor for prognosis after CNSm diagnosis along with aggressive treatment [21]. We demonstrate improved survival following CNSm

diagnosis in the setting of solitary CNSm (median 12.5 months), less than five CNSm (median 6.5 months), absence of extracranial disease (median 14 months), surgical resection of CNSm (median NR) and multimodal therapy for CNSm (median NR). Other studies also support that good performance status at the time of CNSm treatment is significantly associated with improved survival [13, 18]. Given these data, prompt multi-modality treatment approach with neurosurgical and radiation oncology input in the appropriate patient may allow for optimized outcomes following CNSm diagnosis. Overall, this information is valuable to guide patient selection for treatment as well as counseling on expectations and prognosis following CNSm diagnosis.

Although there have been consistent reports that any treatment for CNSm is superior to no treatment, it remains in question whether treatment type affects outcomes independent of disease burden [7]. A survival advantage has been previously reported for patients with CNSm from EC receiving a combination of surgery plus radiotherapy compared to those who did not receive this type of multimodal treatment [21] Unfortunately, studies evaluating treatment type and prognosis have had difficulty discerning whether the aggressive treatment was causally related to improved survival or if the ability to resect CNSm was due to an intrinsic indolence of disease (i.e. small volume) leading to improved outcomes. Due to the close relationship between disease volume and treatment of choice, the present study is met with similar limitations in determining causality of improved outcomes when considering these two variables.

Additional limitations met by this study include a small CNSm cohort and retrospective nature of the study. Although we included patients from two large NCI cancer centers, the total number of patients remains relatively low and as such, the ability for multivariable analysis was limited. We also note that the non-CNSm comparison cohort is quite diverse in that it includes all patients treated for EC over the stated time period. We attempted at add a clinically relevant comparison by selecting out those patients in the non-CNSm cohort who had recurred with similar survival outcomes seen, however this resulted in a significant reduction in number of patients available for comparison. Nonetheless, this study represents one of the largest series in the literature describing CNSm from EC. The patient population was carefully selected to exclude uterine sarcomas to describe outcomes specific to EC more accurately. Population diversity was maintained by including patients from two academic cancer centers with minimal missing data points.

Our data demonstrates that low number of CNSm and multimodal treatment approach most closely predict improved outcomes after CNSm diagnosis. Prompt work-up of new neurologic findings followed by aggressive treatment may be a reasonable treatment option for those with low volume CNSm or minimal extra-CNS disease burden. Given the poor outcomes otherwise, palliative treatment alone and/or hospice care may be considered for EC patients with CNSm who lack the favorable prognostic features identified in this study. Additional molecular profiling of these tumors may allow for elucidation of a mechanism behind the propensity for CNS metastasis in order to aid in early diagnosis and potentially prevention in patients at risk.

## Supporting information

**S1 Data. Available data for central nervous system cohort.**
(XLSX)

**S2 Data. Available data for endometrial cancer without central nervous system cohort.**
(XLSX)

## Author Contributions

**Conceptualization:** Mian M. K. Shahzad.

**Data curation:** Michelle L. Kuznicki, Kristal Ha, Emily Clair McClung, Antonio V. Castaneda.

**Formal analysis:** Biwei Cao, Brooke L. Fridley, Koji Matsuo.

**Investigation:** Michelle L. Kuznicki, Adrianne Mallen, Emily Clair McClung, Antonio V. Castaneda, Biwei Cao, Brooke L. Fridley, Hye Sook Chon, Jing Yi Chern, Mitchel Hoffman, Robert M. Wenham, Koji Matsuo.

**Methodology:** Michelle L. Kuznicki, Adrianne Mallen, Kristal Ha, Emily Clair McClung, Biwei Cao, Brooke L. Fridley, Mian M. K. Shahzad.

**Project administration:** Mian M. K. Shahzad.

**Resources:** Hye Sook Chon, Mitchel Hoffman, Robert M. Wenham.

**Supervision:** Emily Clair McClung.

**Writing – original draft:** Michelle L. Kuznicki, Biwei Cao, Mian M. K. Shahzad.

**Writing – review & editing:** Michelle L. Kuznicki, Adrianne Mallen, Kristal Ha, Emily Clair McClung, Antonio V. Castaneda, Biwei Cao, Brooke L. Fridley, Hye Sook Chon, Jing Yi Chern, Mitchel Hoffman, Robert M. Wenham, Koji Matsuo, Mian M. K. Shahzad.

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
