## [Decision Letter · Decision Letter 0]

23 Mar 2022

PONE-D-21-33642Prognostic Features of Endometrial Cancer Metastasis to the Central Nervous SystemPLOS ONE

Dear Dr. Shahzad,

Thank you for submitting your manuscript to PLOS ONE. After careful consideration, we feel that it has merit but does not fully meet PLOS ONE’s publication criteria as it currently stands. Therefore, we invite you to submit a revised version of the manuscript that addresses the points raised during the review process. We encourage you to address all of the comments made by the reviewers and make changes to the text as indicated by the reviewers. 

We look forward to receiving your revised manuscript.

Kind regards,

Manish S. Patankar, Ph.D.

Academic Editor

PLOS ONE

Journal Requirements:

Additional Editor Comments:

The manuscript has been reviewed by two reviewers. We encourage you to consider the critique carefully and make the necessary changes to the manuscript.

Reviewers' comments:

Reviewer's Responses to Questions

**Comments to the Author**

1. Is the manuscript technically sound, and do the data support the conclusions?

Reviewer #1: Yes

Reviewer #2: Yes

2. Has the statistical analysis been performed appropriately and rigorously? 

Reviewer #1: Yes

Reviewer #2: Yes

3. Have the authors made all data underlying the findings in their manuscript fully available?

Reviewer #1: Yes

Reviewer #2: Yes

4. Is the manuscript presented in an intelligible fashion and written in standard English?

Reviewer #1: Yes

Reviewer #2: Yes

5. Review Comments to the Author

Reviewer #1: Introduction

Line73 : why those two centers ? And how did you decide on those ?

Line74 : can the authors be more specific about the type of outcome they are setting to describe ?

Material and method

Line80-85: few questions :

1- why the years from 1986-2016 for the recurrence ?

2- are those years symmetrical in both institutions ?

3-how did you identify the CNS recurrence in EMR ? Is it through billing of treatments given to those CNS recurrence?

4- how did you match those CNS recurrences with the EC diagnosis? Is it by ICD ?

Line90-93: regarding the control group :

1- why only one center ?

2- why only the years 2014-2017?

3- more importantly, are those control group include patients who have had primary EC without recurrences ? Or is it all EC with recurrences but not CNS met ? Or a mixture of both ( I mean , those with recurrences mixed with those without) ?

Line96: you mentioned that because of he small number of patients only univariate analysis are done , yet in line 102 you mentioned that OS with adjustments for clinical variables . Can you please clarify this discrepancy ?

Line 103-109: are the authors trying to do literature review also ? If so can they put the literature review an an objective ?

Results

Table -1

Looking at the initial stage , we find that majority of cancers are initially are stage 3-4 , which is contrary to the national data ? Can the authors describe this discrepancy or phenomenon ?

If those patients ( who has CNS met ) are initially found to be advanced stage , then a question can be asked if those patients have already had CNS met and it just became a matter of time to surface clinically ?

This can Cleary be seen in table -2 where the discrepancy of the initial stage is clear .

Which brings us to the initial question posted in the material and method about the matter of the control group . It seams that the control group are all EV patients . Which by itself create a huge imbalance between those two groups . And advise is to select the control group to be those EC who had recurrences without CNS met .

Line 131:

The authors mentioned that the majority of patients had syndromes of CNS recurrences, so does that mean that patients are screened for CNS recurrences?

Line 158: the OS is very short for CNS 9 m compared to 49 for EC. Again this is a major discrepancy between the two groups . I would advise the authors to make the comparison to EC who had recurrences without CNS met

Line 161-172: this is good data. I would advise the authors to put this data in a table format for the reader to see the number of patients ..etc

Line 180-181 : the authors here states that HR of death is higher in CNS met : two questions :

1- in retrospective studies , traditionally we analyze the Odd ratio of death ( as Hazard ratio is soared to prospective data) , can the authors deliberately explain please ?

2- the authors mentioned that those HR is after controlling for factor , but the number of patients are too small and the authors said in their material and method that the small number will not allow them to control and be only doing univariate analysis, can the authors please explain this ?

Discussion

Line 262 : the authors mentioned here that they excluded patients with sarcoma, but in table 1 they have sarcoma as a histology . That patient with sarcoma need to be excluded from analysis .

Line 265-266: how is your data super early work up of neurological findings , did the authors study those who did not have an early work up ?

I advise the authors to explore the limitations and the shortcomings of their study and not only discuss its strength

Reviewer #2: Kuzniki et al have reviewed the data from two institutions over three year period from 2014 – 2017. Overall the manuscript is well written and adds valuable information about the incidence, risk-factors and outcomes of a rare event – CNS metastasis in Endometrial Cancer. Information is timely, important as well as well-presented.

Can the authors address the following minor issues to improve the manuscript –

1. More than 50% of the patients are listed as “other” race from the USC cohort. Is there anyway to figure out the actual race/ethnicity in these patients?

2. Table 1 mentions that 36.4% of the patients with CNSm, the first site of recurrence was actually the brain. I am not sure if this is highlighted somewhere else in the manuscript. I think its an important clinical finding that should be highlighted if possible in the results and discussion.

3. Line 145 mentions that only 4.5% patients had leptomeningeal spread. Again, I think its important to highlight in discussion that some patients

4. Please clarify the lines – 161 and 162: Do the authors mean that the survival was better for patients with recurrence site as CNS – but with solitary nodules? I cannot imagine that the patients with multiple CNS Mets did well even if CNS was the only site of Mets.

5. Lines 185-187 are very confusing please clarify

6. Line 215-216 is a better choice for the conclusion of the abstract. Consider using that instead of the one used right now.

7. Table 3 is nicely done but not discussed anywhere except line 232 as a passing comment. Authors should consider comparing their study to published literature.

Congratulation on achieving the difficult task of pooling data across two institutions and thank you for submitting this important study to PLOS One.

6. PLOS authors have the option to publish the peer review history of their article (what does this mean?). If published, this will include your full peer review and any attached files.

Reviewer #1: No

Reviewer #2: No

---

## [Author Response · Author response to Decision Letter 0]

23 Apr 2022

Thank you for the opportunity to re-submit our revised manuscript entitled “Prognostic Features of Endometrial Cancer Metastasis to the Central Nervous System.” We appreciate the reviewers’ constructive feedback. Please find our point-by-point response to reviewer comments and recommended manuscript changes. Reviewers’ comments were very encouraging, and we hope this improved manuscript will be considered for publication in PLOS ONE. All de-identified patient data has also been submitted to PLOS ONE site with this submission.

Reviewer #1: 

Introduction

Line73: why those two centers? And how did you decide on those?

- Author Response: We thank the reviewer for this comment. These sites were chosen as they are the practicing centers of two collaborating senior authors. 

-Manuscript edit: Lines 93-94: Institutions were chosen due to preexisting collaborative relationships and were the main practicing sites of involved authors. 

Line74: can the authors be more specific about the type of outcome they are setting to describe?

- Author Response: We thank you for this comment and opportunity to elaborate. We sought to describe survival outcomes (reported as overall survival from original endometrial cancer diagnosis and survival following diagnosis of central nervous system metastasis.) We have added clarification within the manuscript as written below. 

- Manuscript edit: Line 86-90: “In the present retrospective cohort study, conducted at two major NCI designated cancer centers, we sought to describe characteristics and survival outcomes of patients with CNSm from EC. Additionally, we aimed to evaluate potential prognostic factors associated with improved outcomes following CNSm diagnosis. Due to the rare nature of this presentation, we further aimed to complete a review of the literature and provide additional valuable insights on its outcomes .”

Material and Method

Line80-85: few questions:

1- why the years from 1986-2016 for the recurrence? (see response to #2)

2- are those years symmetrical in both institutions?

- Author Response: Thank you for allowing us to elaborate. The beginning time point 1986 was the earliest year that we were able to find scanned documents for patients reflecting medical evaluation and imaging results to evaluate for central nervous system metastasis. The end time point 2016 was the year when our data collection for this study began at Moffitt Cancer Center. With the addition of patients from USC the search dates were matched to those used at Moffitt for consistency.

- Manuscript edit: Line 96-99: “Electronic medical record systems then were queried at both institutions to identify cases of EC with CNSm diagnosed between January 1,1986 (earliest date of scanned records) through June 1, 2016 (initial start date of data collection).”

3-how did you identify the CNS recurrence in EMR ? Is it through billing of treatments given to those CNS recurrence? (see response to #4)

4- how did you match those CNS recurrences with the EC diagnosis? Is it by ICD ?

- Author Response: We thank the reviewer for this comment and have added appropriate clarification in methods. Patients of interest were identified by selecting patients who had both a diagnosis code for endometrial cancer and a billing code for central nervous system imaging (i.e. brain or spine CT or MRI). 

- Manuscript edit: Line 99-100: “Patients with a diagnosis code of EC and billing code for CNS imaging comprised the screening pool.”

Line90-93: regarding the control group :

1- why only one center ? (see answer to #2)

2- why only the years 2014-2017?

- Author Response: The authors thank the reviewer for this comment. The control group was extracted from a preexisting endometrial cancer database at Moffitt cancer center which encompassed patients treated from 2014-2017, this database only included patients from a single center. Our manuscript now reflects this clarification.

-Manuscript edits: Lines 108-110 “Data from a preexisting database of all EC patients treated at the primary site (MCC) from 2014-2017 was used to create a comparison cohort (non-CNSm).”

3- more importantly, are those control group include patients who have had primary EC without recurrences? Or is it all EC with recurrences but not CNS met? Or a mixture of both (I mean , those with recurrences mixed with those without) ?

- Author Response: We thank you for this thoughtful comment. The control group consists of all patients with EC treated at MCC from 2014-2017 including those both with and without recurrences. Patients with CNSm were not specifically filtered out of this group as the medical record numbers of the patients in the control group were not provided to the study team from the database. Given that patients presented with CNSm both at initial endometrial cancer diagnosis and at the time of recurrence, we felt it was clinically appropriate to compare the CNSm cohort to all patients with an EC diagnosis, regardless of whether they experienced a recurrence. In response to this comment and we included an additional comparison between the CNSm cohort and recurrent patients from the non-CNSm cohort. 

- Manuscript edits: 

- Line 110-111 “This data set included all stages of EC, all EC histologies and were included regardless of recurrence status.”

 - Line 149-156: “An additional comparison was completed between the CNSm cohort and patients within the non-CNSm cohort who developed recurrences (n=30), in attempt to capture the more aggressive cases. In this comparison, the CNSm cohort and the recurrent non-CNSm cohort had similar rates of serous histology (22.7% CNSm vs 23.3% recurrent non-CNSm, p=0.205) and grade 3 tumors (81.8% CNSm vs 63.3% recurrent non-CNSm, p=0.370). However, CNSm cohort continued to have younger age at diagnosis (58.5 years CNSm vs 62.5 years recurrent non-CNSm) and higher rates of stage 3 or 4 disease at time of diagnosis (86.4% CNSm vs 56.6% recurrent non-CNSm, p=0.029.”

 - Lines 207-211: “CNSm cohort OS was also compared specifically to patients within the non-CNSm EC cohort who recurred. Although the CNSm cohort trended towards higher risk of death, this was not significant compared to the recurrent patients from non-CNSm cohort (HR 2.28, 95% CI 0.578-8.974, p=0.239).”

 - Lines 278-280: “Additionally, compared to recurrent non-CNSm EC patients, we were unable to demonstrate a significant difference in survival for CNSm patients (HR 2.28, 95% CI 0.578-8.974, p=0.239)”

Line96: you mentioned that because of the small number of patients only univariate analysis are done, yet in line 102 you mentioned that OS with adjustments for clinical variables. Can you please clarify this discrepancy?

- Author response: Thank you for allowing us to clarify our methods of analysis. For clarity, univariate analysis was completed for comparisons within the CNS metastasis cohort (i.e. figure 1 and 2). Due to small numbers, comparisons within the CNSm cohort used only KM curves and log rank test, without adjustments. Alternatively, in comparing the CNSm cohort to the larger non-CNSm EC cohort (as referenced in lines 180-181), a multivariable cox model was used with adjusting for stage, histology or age. 

- Manuscript edit: Line 120-125 : “For these analyses within the CNSm cohort we did not adjust for covariates due to the limited sample size. For analysis between CNSm and non-CNSm cohorts, Cox proportional hazards regression model was used to estimate hazard ratios (95% confidence interval) for OS with adjustment for clinical covariates. For the analysis using the larger non-CNSm EC cohort, which involved a larger sample size, we were able to fit multivariable models adjusting for stage, histology and age.”

Line 103-109: are the authors trying to do literature review also? If so can they put the literature review an objective?

- Author Response: Thank you for this feedback. We have added this as an aim in the last paragraph of our introduction to better describe our intentions. 

- Manuscript edit: Lines 89-90 : “Due to the rare nature of this presentation, we further aimed to complete a review of the literature and provide additional valuable insights on it outcomes .”

Results

Table -1

Looking at the initial stage , we find that majority of cancers are initially are stage 3-4 , which is contrary to the national data ? Can the authors describe this discrepancy or phenomenon ? If those patients ( who has CNS met ) are initially found to be advanced stage , then a question can be asked if those patients have already had CNS met and it just became a matter of time to surface clinically ? This can Cleary be seen in table -2 where the discrepancy of the initial stage is clear . Which brings us to the initial question posted in the material and method about the matter of the control group . It seams that the control group are all EV patients . Which by itself create a huge imbalance between those two groups . And advise is to select the control group to be those EC who had recurrences without CNS met .

- Author Response: Thank you for this important question and suggestion regarding both table 1 and table 2 highlighting the striking differences between patients who develop CNS metastasis from endometrial cancer and the general endometrial cancer population. As further elaborated in lines 223-227, in our review of the literature, it was found that most patients reported to develop CNSm had an advanced stage endometrial cancer. This likely relates to the intrinsic aggressive biology of endometrial cancers that eventually find their way to the central nervous system. We do concede that our comparison group includes a diverse group of patients with EC including those both with and without recurrences, and therefore our comparison group does not fully address clinical differences in those who recur in the CNSm compared to those that recur elsewhere. Therefore, we have now added a comparison between CNSm and recurrent patients in the non-CNSm cohort. 

- Manuscript edit: Lines 304-308. “We also note that the non-CNSm comparison cohort is quite diverse in that it includes all patients treated for EC over the stated time period. We attempted at add a clinically relevant comparison by selecting out those patients in the non-CNSm cohort who had recurred with similar survival outcomes seen, however this resulted a significant reduction in number of patients available for comparison.”

 - Line 149-156: “An additional comparison was completed between the CNSm cohort and patients within the non-CNSm cohort who developed recurrences (n=30), in attempt to capture the more aggressive cases. In this comparison, the CNSm cohort and the recurrent non-CNSm cohort had similar rates of serous histology (22.7% CNSm vs 23.3% recurrent non-CNSm, p=0.205) and grade 3 tumors (81.8% CNSm vs 63.3% recurrent non-CNSm, p=0.370). However, CNSm cohort continued to have younger age at diagnosis (58.5 years CNSm vs 62.5 years recurrent non-CNSm) and higher rates of stage 3 or 4 disease at time of diagnosis (86.4% CNSm vs 56.6% recurrent non-CNSm, p=0.029.”

 - Lines 207-211: “CNSm cohort OS was also compared specifically to patients within the non-CNSm EC cohort who recurred. Although the CNSm cohort trended towards higher risk of death, this was not significant compared to the recurrent patients from non-CNSm cohort (HR 2.28, 95% CI 0.578-8.974, p=0.239).”

 - Lines 278-280: “Additionally, compared to recurrent non-CNSm EC patients, we were unable to demonstrate a significant difference in survival for CNSm patients (HR 2.28, 95% CI 0.578-8.974, p=0.239)”

Line 131:

The authors mentioned that the majority of patients had syndromes of CNS recurrences, so does that mean that patients are screened for CNS recurrences?

- Author Response: Thank you for the opportunity to clarify this important point. Routine screening for CNS recurrence is not practiced at either institution included in this study. A new symptom preceding CNS imaging was confirmed in 82% of patients. This is now further clarified in our manuscript.

- Manuscript edits: Line 160-162: “Routine CNS imaging to screen for CNSm during treatment or surveillance of EC was not a regular practice at either institution in the absence of symptoms”

Line 158: the OS is very short for CNS 9 m compared to 49 for EC. Again this is a major discrepancy between the two groups. I would advise the authors to make the comparison to EC who had recurrences without CNS met. We thank the reviewer for this important comment. We have added a comparison analysis between CNSm and recurrent non-CNSm EC patients as described above. 

Line 161-172: this is good data. I would advise the authors to put this data in a table format for the reader to see the number of patients ..etc

- Author Response : We appreciate reviewer’s positive comment and agree this is an important result to highlight. As suggested, we have added Table 3 to summarize the data from this portion of the manuscript that is not depicted in Figure 1. 

-Manuscript edit: 202-203: addition of Table 3 

Table 3. Differences in Overall Survival Based on Number of CNSm and Disease Burden

Comparison Groups Median Overall Survival P Value

CNSm only site of metastasis

CNSm + other sites of metastasis 83 months

30 months 0.007

Solitary CNSm

Multiple CNSm 83 months

28 months 0.040

<5 CNSm

≥5 CNSm 49 months

23 months 0.005

Abbreviations: CNSm : central nervous system metastasis, +: in addition to, < : less than, ≥ :greater than or equal to

Line 180-181 : the authors here states that HR of death is higher in CNS met : two questions :

1- in retrospective studies , traditionally we analyze the Odd ratio of death ( as Hazard ratio is soared to prospective data) , can the authors deliberately explain please ?

- Author response: We thank the reviewer for this comment. For the analysis of this study we chose to use a time to event analysis framework (i.e., Cox proportional hazards model). This analysis approach can be used for both retrospective studies and prospective studies in which the outcome of interest is a time to event variable in which some subjects were censored at a particular point in time. Hence, the results are presented in terms of hazard ratios instead of odds ratios, which is the parameter that results from a Cox model.

2- the authors mentioned that those HR is after controlling for factor , but the number of patients are too small and the authors said in their material and method that the small number will not allow them to control and be only doing univariate analysis, can the authors please explain this ?

- Author response: Thank you for allowing us to clarify our methods of analysis. For clarity, univariate analysis was completed for comparisons within the CNS metastasis cohort (i.e. figure 1 and 2). Due to small numbers, comparisons within the CNSm cohort used only KM curves and log rank test, without adjustments. Alternatively, in comparing the CNSm cohort to the larger non-CNSm EC cohort (as referenced in lines 180-181), a multivariable cox model was used with adjusting for stage, histology or age. 

- Manuscript edit: Line 120-125 : “For these analyses within the CNSm cohort we did not adjust for covariates due to the limited sample size. For analysis between CNSm and non-CNSm cohorts, Cox proportional hazards regression model was used to estimate hazard ratios (95% confidence interval) for OS with adjustment for clinical covariates. For the analysis using the larger non-CNSm EC cohort, which involved a larger sample size, we were able to fit multivariable models adjusting for stage, histology and age.”

Discussion

Line 262: the authors mentioned here that they excluded patients with sarcoma, but in table 1 they have sarcoma as a histology. That patient with sarcoma need to be excluded from analysis.

- Author response: Thank you for this close review of our tables. Current clinical practice reflects that uterine carcinosarcoma is more related to endometrial carcinomas instead of uterine sarcomas. Uterine carcinosarcomas are therefore staged and treated within endometrial cancer guidelines which is why we have included this patient in our study. 

- Manuscript edit: Lines 101-104: “Only carcinomas arising from the endometrium were included in the present study, which was limited to endometrioid, serous, clear cell or carcinosarcoma. Uterine sarcomas were excluded”

Line 265-266: how is your data super early work up of neurological findings, did the authors study those who did not have an early work up ?

- Author response: We thank the reviewer for thoughtful comment. Within our cohort of patients with CNS metastasis from endometrial cancer, nearly all of them were diagnosed following the development of new neurologic symptoms. Clinically, new neurologic symptoms may also be attributed to side effects from oncologic treatment or functional decline from failure to thrive. Given the rarity of CNSm from EC, it may be overlooked in the differential diagnosis of new symptoms. Unfortunately, untreated CNSm can result in catastrophic hemorrhage or herniation and therefore early diagnosis and treatment is clinically ideal if a CNSm is present. The purpose of the above statement is to remind the reader that CNSm from EC, although rare, should not be forgotten during evaluation of new symptoms. 

-- Manuscript edit: Lines 266-268: “Identifying CNSm as early as possible is clinically ideal in order to expedite treatment and avoid potentially catastrophic consequences of untreated CNSm including hemorrhage or herniation.”

I advise the authors to explore the limitations and the shortcomings of their study and not only discuss its strength

- Author response: We thank you for this important feedback. We have now elaborately described the study limitations as suggested the reviewer.

- Manuscript edit: Lines 298-308: “Due to the close relationship between disease volume and treatment of choice, the present study is met with similar limitations in determining causality of improved outcomes when considering these two variables. Additional limitations met by this study include a small CNSm cohort and retrospective nature of the study. Although we included patients from two large NCI cancer centers, the total number of patients remains relatively low and as such, the ability for multivariable analysis was limited. We also note that the non-CNSm comparison cohort is quite diverse in that it includes all patients treated for EC over the stated time period. We attempted at add a clinically relevant comparison by selecting out those patients in the non-CNSm cohort who had recurred with similar survival outcomes seen, however this resulted a significant reduction in number of patients available for comparison

Reviewer #2: 

Congratulation on achieving the difficult task of pooling data across two institutions and thank you for submitting this important study to PLOS One.

-Authors response: reviewer’s comments were very encouraging, and authors appreciate the constructive feedback that has helped authors further enhance our manuscript.

Reviewer’s comments: Can the authors address the following minor issues to improve the manuscript –

1. More than 50% of the patients are listed as “other” race from the USC cohort. Is there any way to figure out the actual race/ethnicity in these patients?

- Author Response: We thank the reviewer for this comment. Information regarding race/ethnicity was collected from electronic medical records. At times “other” was listed by the patient themselves or an employee collecting information on patient’s behalf without further specification. Unfortunately, this cannot be further elaborated on with the information available to our research team. 

2. Table 1 mentions that 36.4% of the patients with CNSm, the first site of recurrence was actually the brain. I am not sure if this is highlighted somewhere else in the manuscript. I think its an important clinical finding that should be highlighted if possible, in the results and discussion.

- Author Response: Thank you for this thoughtful comment. We would like to highlight this finding in our manuscript and have added discussion regarding these data as stated below. 

- Manuscript edit: Lines 263-266 . “Additionally, 36% of patients in the present study were diagnosed with CNSm at the time of their first recurrence. This suggests CNSm can occur early in the course of disease recurrence and CNS imaging should be completed in the setting of new neurologic symptoms regardless of timing from EC diagnosis.” 

3. Line 145 mentions that only 4.5% patients had leptomeningeal spread. Again, I think its important to highlight in discussion that some patients

- Author Response: The authors thank you for this feedback and have added discussion below to highlight this important point.

- Manuscript edits Line 268-272: “Interestingly, only one patient in the present study was identified as having leptomeningeal metastasis in the absence of a parenchymal lesion. If neurologic symptoms persist in the absence of gross parenchymal metastasis on brain imaging, a consideration may be made for addition of spinal imaging if this has not been completed.”

4. Please clarify the lines – 161 and 162: Do the authors mean that the survival was better for patients with recurrence site as CNS – but with solitary nodules? I cannot imagine that the patients with multiple CNS Mets did well even if CNS was the only site of Mets.

- Author Response: We thank you for this opportunity to better demonstrate our data. We have added table 3 in addition to the existing figure 1 so that the reader can better visualize our comparison groups. Survival was improved for those with solitary CNS metastasis compared to multiple CNS metastasis. Survival was also improved if CNS was the only site of disease compared to multiple sites of disease. Although this survival information may seem obvious to a clinician, we feel the publication of this survival data may help oncologists select aggressive treatment for a subset of patients who may do surprisingly well despite CNSm diagnosis. As per your suggestions, follow changes have been made to the manuscript:

- Manuscript edits: Addition of Table 3, as well as edits to Lines 190-202: “We identified improved OS when CNS was the only site of metastasis compared to those with additional non- CNS metastasis (83m CNSm only vs 30m additional sites, p= 0.007) (Table 3). Those with CNSm as only site of disease also had improved survival following CNSm diagnosis compared to those with concurrent disease outside CNS (p=0.0096) (Figure 1.1). Out of seven patients with CNSm and no evidence of disease outside of the CNS, only one died during the follow-up period. Those with solitary CNSm had improved OS compared to those with multiple CNSm (83m solitary vs 28m multiple, p=0.04) (Table 3). Patients with solitary CNSm also had improved survival following CNSm diagnosis (p=0.0034) (Figure 1.2). Overall, small number of CNSm was associated with improved OS. OS for patients with less than five CNSm was more favorable compared to those with five or greater CNSm (49m <5 vs. 23m ≥ 5, p=0.0045)(Table 3). Similar survival benefit was noted following CNSm diagnosis (Figure 1.3). No associated significant survival benefit was found with increasing diameter of CNSm (6.5months <3cm vs 14months >3cm, p=0.327).

5. Lines 185-187 are very confusing please clarify : 

- Author Response : Thank you for suggesting to clarify. We were describing outcomes for patients with CNSm from EC that received no further treatment after CNSm diagnosis which appeared confusing. Now, we have re-written these statements to further clarify as recommended. 

- Manuscript edits: Lines 213-215: “Three patients received no treatment for their CNSm, all of whom survived ≤1 month following CNSm diagnosis. Overall survival from EC diagnosis was 29 months, 5 months and 1 month for these three patients”

6. Line 215-216 is a better choice for the conclusion of the abstract. Consider using that instead of the one used right now.

- Author Response : We thank you for this feedback and have redirected this comment to the conclusion paragraph of the manuscript to reflect reviewer comment as suggested. 

- Manuscript edits: Lines 313-314: . “Our data demonstrates that low number of CNSm and multimodal treatment approach most closely predicts improved outcomes after CNSm diagnosis.”

7. Table 3 is nicely done but not discussed anywhere except line 232 as a passing comment. Authors should consider comparing their study to published literature.

- Author response: We thank the reviewer for this comment. Throughout the discussion we have cited most references from Table 3 (now table 4 in revised manuscript) to compare results from the present study to previously published data. Associated reference numbers have been listed in Table 4 to show how they are referenced within the discussion. Thank you again for highlighting this and hope this further clarifies.

---

## [Decision Letter · Decision Letter 1]

9 May 2022

Prognostic Features of Endometrial Cancer Metastasis to the Central Nervous System

PONE-D-21-33642R1

Dear Dr. Shahzad,

We’re pleased to inform you that your manuscript has been judged scientifically suitable for publication and will be formally accepted for publication once it meets all outstanding technical requirements.

Kind regards,

Manish S. Patankar, Ph.D.

Academic Editor

PLOS ONE

Additional Editor Comments (optional):

Reviewers' comments:

Reviewer's Responses to Questions

**Comments to the Author**

1. If the authors have adequately addressed your comments raised in a previous round of review and you feel that this manuscript is now acceptable for publication, you may indicate that here to bypass the “Comments to the Author” section, enter your conflict of interest statement in the “Confidential to Editor” section, and submit your "Accept" recommendation.

Reviewer #1: All comments have been addressed

Reviewer #2: All comments have been addressed

2. Is the manuscript technically sound, and do the data support the conclusions?

Reviewer #1: Yes

Reviewer #2: Yes

3. Has the statistical analysis been performed appropriately and rigorously? 

Reviewer #1: Yes

Reviewer #2: Yes

4. Have the authors made all data underlying the findings in their manuscript fully available?

Reviewer #1: Yes

Reviewer #2: Yes

5. Is the manuscript presented in an intelligible fashion and written in standard English?

Reviewer #1: Yes

Reviewer #2: Yes

6. Review Comments to the Author

Reviewer #1: Thank you for Addressing all questions. Manuscript is ready for publications from my point of view

Reviewer #2: Great job making the manuscript better. The authors have adequately addressed the reviewer comments as well as the statistical comments

7. PLOS authors have the option to publish the peer review history of their article (what does this mean?). If published, this will include your full peer review and any attached files.

Reviewer #1: No

Reviewer #2: No

---

## [Editor Report · Acceptance letter]

17 Aug 2022

PONE-D-21-33642R1 

Prognostic features of endometrial cancer metastasis to the central nervous system 

Dear Dr. Shahzad:

I'm pleased to inform you that your manuscript has been deemed suitable for publication in PLOS ONE. Congratulations! Your manuscript is now with our production department. 

Kind regards, 

on behalf of

Dr. Manish S. Patankar 

Academic Editor

PLOS ONE